# Distinct CholinomiR Blood Cell Signature as a Potential Modulator of the Cholinergic System in Women with Fibromyalgia Syndrome

**DOI:** 10.3390/cells11081276

**Published:** 2022-04-09

**Authors:** Christoph Erbacher, Shani Vaknine, Gilli Moshitzky, Sebastian Lobentanzer, Lina Eisenberg, Dimitar Evdokimov, Claudia Sommer, David S. Greenberg, Hermona Soreq, Nurcan Üçeyler

**Affiliations:** 1Department of Neurology, University of Würzburg, 97080 Würzburg, Germany; erbacher_c@ukw.de (C.E.); d_evdokimov@yahoo.de (D.E.); sommer@uni-wuerzburg.de (C.S.); 2The Edmond & Lily Safra Center for Brain Sciences, The Hebrew University of Jerusalem, Jerusalem 9190401, Israel; shani.vaknine@mail.huji.ac.il (S.V.); gilli.moshitzky@mail.huji.ac.il (G.M.); lina.eisenberg@mail.huji.ac.il (L.E.); david.greenberg1@mail.huji.ac.il (D.S.G.); hermona.soreq@mail.huji.ac.il (H.S.); 3The Alexander Silberman Institute of Life Sciences, The Hebrew University of Jerusalem, Jerusalem 9190401, Israel; 4Department of Pharmacology, College of Pharmacy, Goethe University, Max-von-Laue-Str. 9, 60438 Frankfurt am Main, Germany; sebastian.lobentanzer@uni-heidelberg.de

**Keywords:** fibromyalgia syndrome, cholinergic system, CholinomiRs, microRNA, miR-182-5p, Parkinson’s disease

## Abstract

Fibromyalgia syndrome (FMS) is a heterogeneous chronic pain syndrome characterized by musculoskeletal pain and other key co-morbidities including fatigue and a depressed mood. FMS involves altered functioning of the central and peripheral nervous system (CNS, PNS) and immune system, but the specific molecular pathophysiology remains unclear. Anti-cholinergic treatment is effective in FMS patient subgroups, and cholinergic signaling is a strong modulator of CNS and PNS immune processes. Therefore, we used whole blood small RNA-sequencing of female FMS patients and healthy controls to profile microRNA regulators of cholinergic transcripts (CholinomiRs). We compared microRNA profiles with those from Parkinson’s disease (PD) patients with pain as disease controls. We validated the sequencing results with quantitative real-time PCR (qRT-PCR) and identified cholinergic targets. Further, we measured serum cholinesterase activity in FMS patients and healthy controls. Small RNA-sequencing revealed FMS-specific changes in 19 CholinomiRs compared to healthy controls and PD patients. qRT-PCR validated miR-182-5p upregulation, distinguishing FMS patients from healthy controls. mRNA targets of CholinomiRs bone morphogenic protein receptor 2 and interleukin 6 signal transducer were downregulated. Serum acetylcholinesterase levels and cholinesterase activity in FMS patients were unchanged. Our findings identified an FMS-specific CholinomiR signature in whole blood, modulating immune-related gene expression.

## 1. Introduction

Fibromyalgia syndrome (FMS) is a clinically heterogeneous chronic pain syndrome with a high global prevalence of approximately 2.7% [1,2]. FMS is frequently associated with somatic and mental co-morbidities accompanying predominantly musculoskeletal pain [3]. Adult women are primarily affected, but men may be underdiagnosed [4]. The pathophysiology of FMS remains incompletely understood, and no reliable biomarkers are available, resulting in insufficient diagnostics and treatment [5]. Systemic pro-inflammatory profiles [6,7] and (sub-)clinical autoimmunity [8,9] are among the etiological factors discussed. FMS may aggregate in families and single nucleotide polymorphisms, e.g., in serotonin transporter genes (SLC64A4), the transient receptor potential vanilloid channel 2 gene (TRPV2), or mitochondrial genes, which have been associated with FMS symptoms; however, overall small sample sizes hamper conclusions on an individual patient basis [10,11]. Immune dysregulation based on genetic risk factors, e.g., in mitochondrial genes, may trigger peripheral neuro-inflammatory processes, which, in turn, can induce central and peripheral pain symptoms, and the process may be further enhanced by positive feedback loops between these components [12]. Of note, small nerve fiber degeneration is a common feature in FMS [13,14].

Regarding the autonomic nervous system, higher sympathetic and lower parasympathetic activity was proposed in FMS [15]. Accordingly, FMS patients often suffer from symptoms such as constipation or irritable bladder or bowel syndrome [3,16,17]. A major cholinergic regulator of parasympathetic function is the vagus nerve (VN), which innervates smooth muscles, the gastro-intestinal tract, the heart, and, importantly, immune cells within the spleen [18]. The VN interlinks local and systemic immune processes between the CNS and PNS [19,20,21], and immune cells also express all of the necessary components of the cholinergic machinery [22]. Together, cholinergic mechanisms play an important role in the regulation of cytokines and antibody production [23,24], thereby modulating inflammatory processes with potential implications for FMS pathophysiology and severity [25,26].

In line with the clinical heterogeneity of FMS, experimental assessment of immune correlates such as cytokine transcription or protein levels provide highly variable data, which often leads to conflicting results between different study cohorts [27,28]. Here, we investigated microRNAs (miRs) expression patterns in FMS, since these small RNAs act as posttranscriptional regulators in health and disease [29,30]. Importantly, miRs from blood can be easily obtained, serving as potential biomarkers, but also represent direct modulators in pain conditions [31]. We focused specifically on CholinomiRs, a class of miRs regulating the cholinergic system. Each CholinomiR may target five or more cholinergic mRNA transcripts, thereby modulating key processes that underlie inflammation, physiological stress, and anxiety [32,33,34].

We hypothesized that the dysregulation of the systemic cholinergic system is a potential causal element of FMS pathophysiology with an impact on inflammation processes in these patients. To test this hypothesis, we carried out small RNA-sequencing (RNA-seq) and quantitative real-time PCR (qRT-PCR) validation, followed by cholinergic mRNA target analysis and cholinesterase activity measurements in blood samples from FMS patients in comparison to healthy controls. Further, we compared the resultant FMS miR signature to those of patients suffering from Parkinson’s disease (PD)-related pain as disease controls.

## 2. Materials and Methods

### 2.1. Study Participants and Biomaterial Collection

Figure 1 provides a synopsis of all study participants, the biomaterial used, and the experiments performed. Forty-nine female FMS patients and twenty-five age- and sex-matched healthy controls were part of a previously characterized cohort [35]. Briefly, FMS patients were recruited according to the 1990 and 2010 diagnostic criteria of the American College of Rheumatology (ACR), and all participants underwent complete neurological examination at the Department of Neurology, University of Würzburg. The study was approved by the Ethics Committee of the University of Würzburg Medical Faculty (#121/14), and all study participants gave written informed consent. For assessment of pain levels and pain-related impairment, the Graded Chronic Pain Scale (GCPS) was applied [36]. Mean pain intensity within the last six months was recorded on an 11-point numeric rating scale (NRS) with 0 = no pain and 10 = worst pain, defined as NRS_Pain_. Furthermore, patients were stratified for their subjective degree of pain-associated impairment of daily life activities, which were also rated on the NRS with 0 = normal life activities and 10 = no daily life activities possible, defined as NRS_Daily life_. Hereby, we defined NRS_Daily life_ 0–2 as “mildly impaired” and NRS 3–10 as “severely impaired”.

Tempus™ Blood RNA Tubes (Thermo Fisher Scientific, Waltham, MA, USA) were used to collect 3 mL of whole blood from FMS patients and matched healthy controls. Blood was withdrawn in the morning with participants restrained from extensive physical activity and heavy meals the prior day, after overnight fasting. Cells were lysed and RNA was stabilized via 30 s of rigorous shaking of the tube. RNA was either extracted on the same day, or tubes were stored at −20 °C until extraction. The MagMAX™ for Stabilized Blood Tubes RNA Isolation Kit (Thermo Fisher Scientific, Waltham, MA, USA) was used for total RNA extraction according to the manufacturer’s protocol. RNA quality and quantity were assessed with a NanoDrop™ One (Thermo Fisher Scientific, Waltham, MA, USA), and RNA was stored at −80 °C. Sequenced samples were analyzed via an Agilent 2100 Bioanalyzer system (Agilent Technologies, Santa Clara, CA, USA; see details for RNA quality in Appendix A).

Regarding the miR signature control groups, we investigated sequencing data from the whole blood of women with PD and matched healthy controls, obtained from the Progression Markers Initiative (PPMI) database (https://www.ppmi-info.org/accessdata-specimens/download-data (accessed on 1 March 2021); updated information on the PPMI study is available at www.ppmi-info.org). In order to compare the PD cohort to our FMS and control cohort results, we only used blood samples drawn from female PD patients and controls who matched the following criteria: (1) Samples were taken at the baseline time point of the study, (2) patients were not taking any PD-related medication, (3) their age was ≤65 years, and (4) participants were from idiopathic (sporadic) PD and the control cohorts only. The Unified Parkinson Disease Rating Scale (UPDRS) was applied, and the item 1.9 pain and other sensations were utilized as NRS_Pain impairment_, with NRS = 0–1 as no pain-related impairment and NRS = 2–4 as pain-related impairment.

PPMI provided small RNA-sequencing data from whole blood samples of PD patients and of matched healthy controls that were collected by venous draws in PAXgene Vacutainer tubes (Qiagen, Hilden, Germany). These were incubated at room temperature (18–25 °C) for 24 h before final storage at −80 °C. RNA was extracted using the PAXgene blood miRNA Kit protocol (Qiagen, Hilden, Germany; see details for RNA quality in Appendix A).

### 2.2. Small RNA-Sequencing and miR Analysis

We subjected a randomized subset of 31 FMS and 17 control blood samples to small RNA-sequencing. For this purpose, 300 ng RNA per sample served for library preparation using the NEB-Next^®^ Multiplex Small RNA Library Prep Kit for Illumina^®^ (Index Primers 1-48; NEB-E7560S; New England Biolabs, Ipswich, MA, USA) at the National Center for Genomic Technologies at the Hebrew University of Jerusalem. According to the information provided by the manufacturer, this kit contains a size selection step after PCR to ensure that only small RNAs up to 140 nt length are sequenced. Next generation short RNA-sequencing was conducted on an Illumina NextSeq 500, using two Illumina NextSeq 500/550 High Output Kit v2.5 (75 Cycles) flow cells (20024906), with 24 samples each (all Illumina, San Diego, CA, USA). Quality control was performed using FastQC, version 0.11.8 [37]. Short RNA was aligned to the miRBase version 21 using miRExpress 2.0 with default parameters with no mismatches allowed [38,39]. We defined a miR as “CholinomiR”, if five or more cholinergic genes assessed by the previously published miR-targeting graph database (‘miRNet’) [40] were predicted targets of the respective miR. First, a minimum threshold of seven was applied for the prediction algorithm to define a targeting relationship towards a gene transcript and every DE miR. In the second step, miRs targeting more than four transcripts of a defined cholinergic gene list were termed CholinomiRs. To assess whether the found CholinomiRs were enriched in dysregulated FMS miRs, we compared their fraction against the fraction of detected, but not DE, miRs via Fisher’s exact test. The Cytoscape software (v.3.9.0) was used for visualization of the miR-gene target connections [41].

Regarding the PD disease control cohort, sequencing data was provided by PPMI. In summary, small RNA was sequenced at Hudson Alpha’s Genomic Services Lab on an Illumina NovaSeq6000 (San Diego, CA, USA). All samples were prepared using the Bioo smRNA library prep Kit (Bioo Scientific Corporation, Austin, TX, USA). Binary base calls were converted to FASTQ’s using Illumina bcltofastq v1.8.4, and FASTQ’s were merged and processed with miRMaster v1.0 [42]. To identify miRNAs, reads were mapped to miRBase v22 precursors with Bowtie 1.1.2 and were processed with miRMaster to allow up to 1 mismatch and 2 nt overlap at the 5′ end and 5 nt overlap at the 3′ end of the miRNA annotation [43,44]. The complete PPMI study information, including sample collection and other clinical protocols, is available at https://www.ppmi-info.org/ (accessed on 1 March 2021).

### 2.3. Validation of miR and mRNA Transcripts

For reverse transcription of miRs, we used the miRCURY LNA RT Kit (Qiagen, Hilden, Germany). A total of 10 ng RNA in 2 µL nuclease-free water was added to a mix containing 2 µL reaction buffer, 5 µL nuclease-free water, and 1 µL enzyme mix. Reactions were performed on a PRISM 7700 Cycler (Applied Biosystems, Waltham, MA, USA) under the following conditions: reverse transcription (42 °C, 60 min) and enzyme deactivation (95 °C, 5 min). For mRNA, TaqMan Reverse Transcription reagents (Thermo Fisher Scientific, Waltham, MA, USA) were used. For each sample, 250 ng mRNA was pre-incubated with 5 µL random hexamer at 85 °C for 3 min, followed by reverse transcription in a reaction mix, containing 10 μL 10× PCR buffer, 6.25 μL multiscribe reverse transcriptase, 2 μL RNase inhibitor, 22 μL MgCl_2_, and 20 μL deoxyribonucleoside triphosphate. Reactions were performed on a PRISM 7700 Cycler (Applied Biosystems, Waltham, MA, USA) under the following conditions: annealing (25 °C, 10 min), reverse transcription (48 °C, 60 min), and enzyme inactivation (95 °C, 5 min). Transcribed cDNA was stored at −20 °C before further analysis.

A QPCR of miR and mRNA targets was carried out on a QuantStudio 3 (Thermo Fisher Scientific, Waltham, MA, USA) using the ΔΔCt method for relative quantification. The miRCURY LNA SYBR Green PCR Kit (Qiagen, Hilden, Germany) and pre-designed miRCURY LNA miR PCR Assays (Qiagen, Hilden, Germany) were applied for each miR (see list of primers in Table 1). Following a dual selection approach, we included ten transcripts as potential endogenous controls for small RNAs. SNORD38B, SNORD44, SNORD48, 5S RNA, and hsa-miR-221-3p were selected from the literature. Furthermore, miRs with a mean base read level of >1000 reads within the small RNA-sequencing dataset were selected by their intra- and intergroup stability, assessed by NormFinder [45], identifying hsa-miR-942-5p, hsa-miR-194-5p, miR-93-5p, miR92a-3p, and miR423-5p as candidates. Twelve random CTR and FMS samples were used for validation via qRT-PCR. SNORD38B, SNORD44, SNORD48, and miR-423-5p were selected as suitable endogenous controls, based on Ct intergroup comparability and on standard deviation and range across all samples. To minimize the risk of a potentially skewed normalization by only one endogenous control, we normalized against the combined geometric mean of these four transcripts. Each well contained 5 µL 2× miRCURY SYBR Green Master Mix with 1 µL ROX per 50 µL, 1 µL primer, and 4 µL of 1:80 diluted cDNA. Functionality and specificity of SYBR green-based primers were checked via a melting curve step assessment. Gene expression analysis was performed with TaqMan qRT-PCR reagents (all Thermo Fisher Scientific, Waltham, MA, USA) with pre-designed Assays using RPL13A as an endogenous control. Each well contained 0.5 µL nuclease free water, 5 µL Fast Advanced Mastermix, 0.5 µL RPL13A primer, 0.5 µL target primer, and 3.5 µL cDNA.

### 2.4. Cholinesterase Activity Assay

From a randomly selected subset of study participants (FMS, n = 29; CTR, n = 11), the cholinergic status in serum samples was analyzed via Ellman’s assay [46]. The thawed serum samples were diluted 1:6 in PBS and incubated with Ellman solution containing 0.1 M sodium phosphate buffer, with pH 7.4 and 0.5 mM 5,5′-dithiobis-(2-nitrobenzoic acid) (DTNB). For acetylcholinesterase (AChE) activity, 10 mM of the specific inhibitor tetraisopropyl pyrophosphoramide (iso-OMPA; Sigma, St. Louis, MO, USA) was added at 1:200 dilution in Ellman’s solution to block butyrylcholinesterase (BChE) activity. Samples were incubated for 90 min at room temperature. After adding the substrate acetylthiocholine (ATCh) in a final concentration of 1 mM, kinetic measurements were performed at 37 °C and 405 nm wavelength using a Tecan Spark microplate reader (Tecan Group, Männedorf, Switzerland). The activities of each sample were measured in triplicates and were repeated once. To diminish potential intra-test differences, each microtiter plate contained an in-house control triplicate of recombinant human AChE and BChE enzymes. Raw data were collected in units of “mean optical density (OD)/minute” and were processed as activity in nmol ATCh hydrolyzed/min/mL according to the Beer–Lambert law (A = εlc; molar absorptivity constant ε 2-nitro-5-thiobenzoate = 13,600), and they were multiplied by 6, accounting for the serum dilution factor.

### 2.5. Statistical Analysis and Visualization

Differential expression (DE) analysis of the transcriptomic profiles from the FMS cohort was performed using the “DESeq2” package via R platform [47,48]. Differences were assumed statistically significant at *p* < 0.05 after false discovery rate (FDR) correction, and data were filtered for a total count > 300 across all samples. DE analysis of the PPMI PD cohort was conducted in the same manner in DESeq2. Counts per million (CPM) measurements for downstream analysis were calculated using the edgeR R package [49,50]. Principal component analysis (PCA) and linear discrimination analysis (LDA) were performed in R using the “stats” “MASS” packages accordingly, as well as the graphic packages “ggfortify”, “ggplot2”, and “ggConvexHull” [48,51,52,53,54]. Briefly, PCA components (PCs) were based on finding the biggest variability between samples, whereas LDA components (LDs) explored the biggest variability between the known groups (FMS_mild_imp, FMS_severe_imp, and control).

For data derived from qRT-PCR, correlations, and Ellman’s assay, SPSS 27 (IBM, Armonk, NY, USA) was used for analysis and was plotted in GraphPad Prism 8 (GraphPad Software, Inc., La Jolla, CA, USA) for visualization. The Mann–Whitney U Test was used for comparison of two groups, and the Kruskal–Wallis test was used for correlations. When appropriate, Holm–Sidak post correction was applied. Furthermore, the open-source vector software Inkscape V 0.92 (https://inkscape.org (accessed on 12 March 2018)) was used, and graphical icons were integrated from Smart Servier Medical Art https://smart.servier.com/ (accessed on 21 December 2021)) under the CC BY 3.0 license for visualization.

## 3. Results

### 3.1. Clinical Characteristics

The main clinical characteristics of the study population are summarized in Table 2.

### 3.2. Identification and Characterization of Systemic CholinomiRs in FMS

Small RNA-seq of women with FMS (n = 31) versus female healthy controls (n = 17) (Figure 2A) revealed 26 down- and 43 up-regulated miRs. Assessment of miR-mRNA targeting relations connected to cholinergic genes [35] identified 19 altered CholinomiRs within the dataset (Figure 2B).

The FMS-modulated CholinomiRs included upregulation of hsa-miR-374a-5p, -9-5p, -182-5p, -548d-5p, -454-3p, -183-5p, -101-3p, -148a-3p, -7-5p, -128-3p, -186-5p, and -27b-3p. Hsa-miR-532-3p, -328-3p, -766-3p, -1275, -625-5p, -671-5p, and -3609 showed downregulation in FMS blood samples (Figure 2B). In general, upregulated CholinomiRs from Tempus-tube-derived RNA showed higher reads compared to downregulated ones (Figure 2C). Since these RNA samples contained mRNA and small RNA from the whole variety of blood cells, we used an open database to assign our CholinomiRs to subtypes of nucleated white blood cells (WBC) and to red blood cells (RBC). Low frequency transcripts referred predominantly to single or few cell types. For instance, hsa-miR-548d-5p was restricted to monocytes, and hsa-miR-3609 was abundant only in CD4+ T cells. Hsa-miR-183-5p was restricted to RBC, but it exhibited one of the highest expression profiles across the sequenced samples (Figure 2B,C). Furthermore, CholinomiRs with stronger upregulation and more copies, reflected in higher counts such as hsa-miR-182-5p, hsa-miR-148a-3p, or hsa-miR-101-3p, were more broadly expressed according to the microRNA catalogue of human peripheral blood (http://134.245.63.235/ikmb-tools/bloodmiRs (accessed on 02 December 2021)). The larger fraction of CholinomiRs was upregulated and also connected to a higher number of cholinergic targets, potentially having more impact on the cholinergic system.

### 3.3. CholinomiR Signature Is Pain-Related and FMS-Specific

A comparison of ratios between detected CholinomiRs within the set of DE (19 of 69; 27.5%) and not DE miRs (68 of 474; 14.3%) revealed enrichment of dysregulated CholinomiRs in FMS, indicating that the observed signature in whole blood RNA is not random (*p* = 0.008; Fishers’s exact test). To evaluate the relevance of our CholinomiR subset for discriminating FMS patients from CTR, we compared the PCA pattern of all DE miRs to a PCA focusing on the 19 FMS-modulated CholinomiRs. The general PCA poorly differentiated both cohorts, and our CholinomiR-specific PCA yielded a slightly improved separation (Figure 3). When stratifying the group for the degree of daily life impairment due to pain in FMS and subjecting three groups to LDA, we achieved better discrimination from controls for all DE miRs as variables. This was further enhanced by applying only CholinomiRs as variables, distinguishing FMS with high daily life impairment without overlap versus the controls. Notably, only one FMS patient with low pain-related daily life impairment dispersed towards controls, and one dispersed towards the firmly impaired FMS subgroup (Figure 3).

Next, we asked if our CholinomiR signature is FMS-specific or if it reflects pain that may accompany other chronic diseases that involve immune-related alterations. For this purpose, we compared our findings with those of small RNA-seq of whole blood samples obtained from female PD patients (n = 92) versus female CTR (n = 47). Notably, 13 out of the 19 CholinomiRs determined in FMS patients were also detected in this analysis, affirming their expression in blood (Figure 4A). However, these CholinomiRs were not dysregulated in PD, and only hsa-miR-128-3p showed a trend towards downregulation (Figure 4B) in contrast to upregulation in FMS.

Pain is a relevant non-motor symptom in PD with significant impacts on patients’ health-related quality of life [56]. Therefore, we applied our FMS CholinomiR signature in blood samples of PD patients as a predictor of pain-related everyday life impairment. Hsa-miR-625-5p showed a trend towards downregulation in PD patients who reported pain-related impairment (Figure 4C), in line with the downregulation observed in FMS. We also correlated CholinomiR expression with FMS patients’ pain levels and found that hsa-miR-27b-3p (r = 0.373; *p* = 0.039) and hsa-miR-148a-3p (r = 0.362; *p* = 0.045) moderately correlated with NRS_Pain_ within the FMS cohort (Figure 4D,E). Taken together, these observations indicate biological relevance of CholinomiR alterations in FMS, distinct from PD patients, and hint towards an association with pain.

### 3.4. CholinomiR Hsa-miR-182-5p Is Upregulated in Women with FMS Compared to Healthy Controls

To experimentally validate our sequencing approach, we quantified hsa-miR-182-5p levels, using four selected small RNA transcripts as endogenous controls (Figure 5A). This broadly expressed CholinomiR showed a relatively strong upregulation (log2fold = 0.76) in small RNA-seq, revealing a twofold upregulation (median increase = 1.93) in FMS patients compared to CTR via qRT-PCR in our entire cohort (Figure 5B).

### 3.5. Female FMS Patients Inherit a Dense CholinomiR-Gene Transcript Network

The previously published miR-targeting graph database ‘miRNet’ allowed in-depth exploration of the CholinomiR-gene transcript network by integrating publicly available validated and predicted miR-gene interaction datasets [40]. Our analysis revealed 47 potential cholinergic targets and 118 miR-mRNA interactions (Figure 6) with upregulated cholinergic miRs, forming 80 connections compared to 38 connections of downregulated miRs (for details see Appendix A).

Correspondingly, the miR-mRNA network showed a collective regulation of a subset of transcripts mostly targeted by upregulated miRs. Examples include bone morphogenetic protein receptor type 2 (BMPR2), linked to ten CholinomiRs (eight upregulated, two downregulated), RAR-related orphan receptor A (RORA) with seven connections (six upregulated, one downregulated), and interleukin 6 cytokine family signal transducer (IL6ST) with five upregulated CholinomiR connections, together indicating general downregulation of these transcripts. In contrast, the cholinergic receptor nicotinic beta 2 subunit (CHRNB2) was a predicted target of only downregulated CholinomiRs. Lastly, acetylcholinesterase (AChE), a key enzyme for cholinergic signaling via breakdown of acetylcholine, was equally linked to three up- and three downregulated CholinomiRs (Figure 6). We compared our candidate gene list with an online mRNA sequencing dataset derived from healthy donor blood RNA, from the Human Protein Atlas (HPA) project [57] (RNA HPA blood cell gene data; https://www.proteinatlas.org/about/download (accessed on 27 February 2022)) to assess the expression profile of these miR targets across single immune cell subtypes (see Appendix A). With 61.7% (29/47 genes), natural killer (NK) cells expressed the highest number of mRNA targets, followed by neutrophils with 57.5% (27/47), whereas myeloid dendritic cells showed the lowest numbers of detected mRNA targets at 44.7% (21/47). Among the top-targeted transcripts, BMPR2, IL6ST, RORA, and NR1D2 were expressed in almost all cell types, whereas ACHE and BDNF expression was not detected in this dataset. However, RBC was not included in the HPA dataset, which represents a large fraction of Tempus-tube-derived RNA, and no disease-associated immune cell data are included.

### 3.6. Downregulation of Highly Targeted Cholinergic Gene Transcripts

To assess the direct influence of CholinomiRs on cholinergic transcript levels, we performed qRT-PCR with eleven associated target genes: BMPR2, IL6ST, RORA, nuclear receptor subfamily 1 group D member 2 (NR1D2), AChE, brain-derived neurotrophic factor (BDNF), nerve growth factor receptor (NGFR), bone morphogenetic protein 3 (BMP3), serine and arginine rich splicing factor 6 (SRSF6), cholinergic receptor nicotinic alpha 4 subunit (CHRNA4), and cholinergic receptor nicotinic beta 2 subunit (CHRNB2). Low numbers of CHRNA4 and CHRNB2 mRNA transcripts in whole blood RNA did not allow comparisons between FMS and CTR via qRT-PCR and were excluded from analysis. Within the group of cholinergic transcripts with five or more associated upregulated CholinomiRs, BMPR2 and IL6ST were downregulated (Figure 7B,C) in FMS, whereas RORA and NR1D2 expression levels were similar between FMS and CTR (Figure 7D,E). None of the other investigated transcripts were altered in FMS compared to CTR (Figure 7F–J).

### 3.7. Unchanged Cholinesterase Activity in FMS

Although not altered at the transcriptional level in FMS, we considered that AChE may be affected in FMS at the protein level, indicated by altered enzyme activity. The availability and concentration of acetylcholine (ACh) as the ligand of cholinergic signaling depends on its degradation by AChE and BChE. Hence, we measured the activity of total cholinesterases as well as AChE and BChE specific activity in serum samples of FMS patients. Notably, FMS and CTR serum exhibited similar levels of both total and AChE- or BChE-specific hydrolytic activity (Figure 8).

## 4. Discussion

We performed a detailed profiling of CholinomiRs in whole blood samples of patients with FMS and compared our data to healthy and disease controls. We found moderate DE of 69 miRs, including 19 cholinergic miRs that distinguished FMS from the investigated control groups and that may modulate downstream gene expression. CholinomiRs play an important role in orchestrating immune-related responses in neurodegenerative diseases, depression, and inflammatory disorders, including inflammatory bowel disease [58,59,60,61]. Only few studies have investigated miR levels in FMS blood so far. Two studies used serum, whereas one report focused on peripheral blood mononuclear cells (PBMC). Their approaches relied on miRNA microarrays with varying panels, a smaller sample size, and a single set of hybridization conditions, compromising the outcome of these tests [62,63,64]. Less than ten DE miRs were identified in each study, without overlap between studies, and none of the CholinomiRs we present here was reported. Hence, our small RNA-seq represents the first comprehensive and unbiased survey of miR expression in FMS.

Mapping our CholinomiRs to blood cell types revealed dysregulation of commonly expressed CholinomiRs, such as hsa-miR-27b-3p, hsa-miR-148a-3p, and hsa-miR-182-5p, but also that of cell-type-specific miRs, such as hsa-miR-3609 in CD4+ T cells or hsa-miR-548d-5p in CD14+ monocytes. CholinomiR dysregulation in FMS was nonrandom, and highly abundant members tended to be upregulated.

Focusing on DE CholinomiRs improved separation between FMS and healthy control groups, especially via LDA and subgrouping between low and high pain-related impairment in daily life. To evaluate whether our CholinomiR signature referred to a general pattern in chronic diseases with suspected immune-related component, we cross-examined expression of our CholinomiRs in female PD patients of comparable age. Intriguingly, FMS presented a strikingly higher and clearer overall DE of CholinomiRs than PD, in which only hsa-miR-128-3p showed a trend towards reduced expression compared to controls, inverse to its upregulation in FMS. Hsa-miR-128-3p is involved in the neuronal oxidative stress response via interaction with the circular RNA circSLC8A1, which blocks its functioning and modulates the process of nonsense-mediated RNA decay [65,66,67]. PD patients with pain-related impairment showed a tendency towards hsa-miR-625-5p downregulation, inverse to its upregulation in FMS, hinting towards opposed pain-related function. In FMS, hsa-miR-148a-3p and 27b-3p positively correlated with pain intensities, indicating a putative association with pain severity. Interestingly, hsa-miR-27b-3p is implicated with inflammatory modulation of toll-like receptors [68]. Although we do not define a “unique” FMS CholinomiR signature, comparisons with healthy and disease controls may indicate distinct small-RNA-mediated mechanisms of pain in FMS and PD. Pain phenotypes, perceptions, and methodological assessments of pain substantially vary across individuals and between diseases. The exact roles of these candidate miRs in different pain conditions needs to be investigated in future studies with respective patient cohorts.

Validating our RNA-seq results, qRT-PCR confirmed the upregulation of hsa-miR-182-5p in FMS. This miR targets, e.g., BDNF, which predominantly derives from the central nervous system, but it is also secreted by activated immune cells and acts as a positive regulator of survival and differentiation of PNS and CNS neurons [69]. Reduced blood levels of BDNF have been reported in major depression (MD), and depressed mood is a frequent co-morbidity in FMS [3,70]. However, contrary to MD, some studies reported increased serum BDNF levels in FMS patients [71,72], whereas others did not find any deregulation [73], which obscures its role in FMS pathology. Our study exclusively focuses on blood RNA expression, leaving our findings hinting towards a potential lack of BDNF deregulation speculative. Hsa-miR-182-5p is also associated with NR1D2, which is a gene involved in circadian rhythmicity and which regulates lipid metabolism and inflammatory responses [74,75]. CholinomiRs also targeted RORA and Clock Circadian Regulators as further members of the circadian rhythm gene family. Intriguingly, circulation of leukocytes oscillates in a diurnal manner, and amplification of the circadian expression of pro-inflammatory mediators is known in chronic diseases [76]. As an example, the circadian regulator cortisol modulates T-cell activity and is assumed to play a role in FMS pathophysiology [77,78,79]. CholinomiR-gene transcript networks predicted further relevant candidates of immune regulation. BMPR2 and several BMPs are involved in NK activation and autoimmunity as part of the TGFβ superfamily [80,81,82]. IL6ST mediates immune responses by binding to other receptors, such as IL-6R, forming high-affinity receptor complexes [83].

We investigated these key mRNA transcripts targeted by CholinomiRs of interest via qRT-PCR in our whole blood samples. Indeed, BMPR2 and IL6ST levels were reduced in FMS patients in accordance with the miRs most prominent role of post-transcriptional repressors. The membrane receptors BMPR2 and IL6ST are pivotal components of the TGFβ-superfamily-mediated SMAD (BMPR2) and the IL-6-family-mediated JAK–STAT (IL6ST) pathways [84]. In general, the activation of IL-6-mediated pathways exerts pro-inflammatory cellular responses [83]. BMP signaling can be bidirectional, dependent on ligand–receptor combination and the respective cell [80,85], however, a link with autoimmunity is suspected, e.g., in multiple sclerosis, and BMPs can promote Th17 proliferation while suppressing T-reg cell generation [86,87]. Although further validation of our data is mandatory by investigating larger patient samples, CholinomiRs may act on BMPR2 and IL6ST in a compensatory manner to confine underlying pro-inflammatory processes in FMS. Investigated circadian transcripts RORA and NR1D2 and transcripts targeted by fewer miRs or connected to both up- and downregulated CholinomiRs, such as AChE and BDNF, were not altered in whole blood. Accordingly, serum AChE activity was sustained in FMS patients compared to controls. Together, this indicates that the observed CholinomiRs modifications may serve to maintain a balanced system.

Traditional screening approaches failed to identify robust pro-inflammatory “signatures” or precise immune-related pathways in FMS [27,28]. However, recent advances with in-depth analyses have revealed immune and neuro-immune mechanisms as potentially involved in FMS pathophysiology. Passive transfer experiments and cell culture incubation along with immunoreaction in tissue sections indicated binding of FMS patient-derived IgG at the dorsal root ganglion level, representing a neuro-inflammatory component via the humoral system [8]. Moreover, a switch from circulating to resident hyper-responsive natural killer (NK) cells in FMS may participate in peripheral neurodegeneration [88]. Both NK activation and autoimmunity are linked to BMP signaling, e.g., via an autocrine activation pathway via BMP receptors and BMP ligands in NK cells [80,81,82], which fits our identified target genes BMP1, BMP2, BMP3, BMP7, BMP6, BMPER, BMPR1B, and, most importantly, BMPR2. Finally, changes in monocyte subtype distribution and amplified p38 MAPK/MK2 axis activity may be associated with FMS [89,90]. In this respect, monocyte-specific hsa-miR-9-5p and globally expressed hsa-miR-101-3p both target JAK2, which is pivotal for IL6/JAK2/STAT3 axis-mediated inflammation. Taken together, CholinomiRs may shift inflammatory processes via modulation of the systemic cholinergic system.

From the clinical point of view, our findings give further evidence for a distinct pathophysiology of FMS and draw attention to its links with the cholinergic system. Indeed, the anticholinergic drug amitriptyline is one of the first-line pharmacological treatment options for FMS patients [91,92]. Although we cannot rule out a potential effect of CholinomiRs on the individual response of FMS patients to anticholinergic medication, since only three (for small RNA-seq) and six (for qRT-PCR) FMS patients were on low dosages of amitriptyline (Table 1), we believe that this aspect deserves consideration in future studies. Given the individually diverse effects and side effects of amitriptyline in FMS, we speculate that a pathophysiological link to distinct CholinomiRs and related pathways may be possible.

## 5. Limitations

Analysis of circulating miRs from serum or plasma can be severely distorted by hemolysis during preparation [93], and little information can be drawn on source or effector cells. Blood sample processing and isolation of PBMC prior to analysis affects gene regulation within hours and may affect miR transcription even more rapidly [94,95,96]. In contrast, our approach allowed us to create a direct snap shot of the transcriptomic state, since all cells were immediately lysed and total RNA was stabilized [97]. However, several limitations must be addressed. Our assignment of miRs towards particular blood cell types relied on a database mapping approach and was not directly assessed. Only FMS patients with normal routine blood assessment results were included [35] to avoid conditions in which aberrant miR expression patterns may be affected by blood cell distributions, but we did not analyze single blood cell type percentages. Finally, expression changes of targets restricted to a single blood cell subtype or transcripts diametrically regulated across blood cells subtypes are likely masked within global blood RNA and can therefore not be identified via qRT-PCR. Experiments with sorted blood cells are necessary to disentangle immune-related mechanisms in the pathophysiology of FMS related to CholinomiRs, which may be particularly relevant for the identified genes involved in circadian rhythmicity.

FMS and control blood was obtained via Tempus tubes, and PD patient and respective control blood collection employed the PAX system. Both systems rely on direct cell lysis and RNA stabilization; nevertheless, a previous study called for caution in cross-evaluation of expression levels between both systems [98]. To avoid fallacious conclusions, we only compared whether global deregulation of CholinomiRs was seen against their respective control cohort.

Importantly, small non-coding RNA regulators also comprise a recently introduced class of tRNA fragments (tRF) with diverse cellular roles, including mRNA repression, that can take over miR functions in a ‘changing of the guards’ manner [60,99]. Hence, to gain insight into the detailed small RNA landscape, tRF analysis needs to be addressed in future studies for both FMS and PD.

## 6. Conclusions

We investigated systemic blood CholinomiR signatures in women with FMS and compared our data with those of healthy controls and patients with PD associated with somatic pain as disease controls. CholinomiRs play an important role in neurodegenerative and inflammatory diseases. In FMS subgroups, degeneration of peripheral small nerve fibers was observed, and inflammatory bowel disease and affective cognitive symptoms were common comorbidities. We found a distinct CholinomiR signature, which distinguishes FMS from healthy subjects and PD patients as disease controls and links these CholinomiRs with downstream immune-related mRNA transcripts. These findings provide evidence for miR regulation of immune cell processes in FMS with an emphasis on the systemic cholinergic system. Although the exact pathophysiological mechanisms remain unclear, CholinomiRs emerge as important post-transcriptional regulators, which may mediate pathological pro-inflammatory or compensatory effects in immune cells of FMS patients.

## Figures and Tables

**Figure 1 cells-11-01276-f001:**
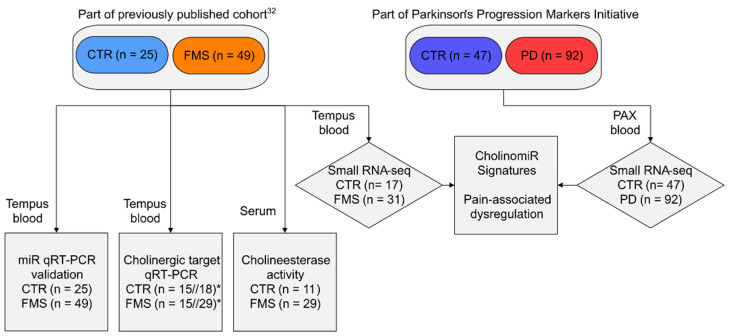
Study workflow. Abbreviations: CTR, healthy control; FMS, fibromyalgia syndrome; PD, Parkinson’s disease; miR, microRNA; qRT-PCR, quantitative real-time PCR. Tempus blood, blood samples collected in Tempus™ Blood RNA Tubes; PAX blood, blood samples collected in PAXgene Vacutainer tubes. * For qRT-PCR either CTR (n = 15) versus FMS (n = 15) or CTR (n = 18) versus FMS (n = 29) samples were used.

**Figure 2 cells-11-01276-f002:**
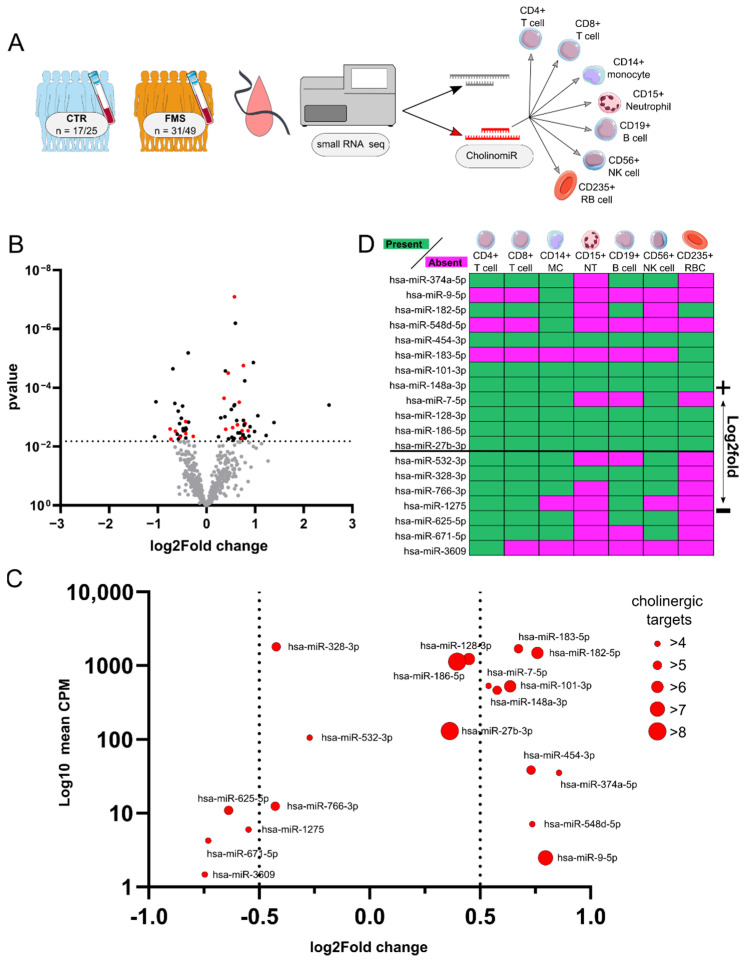
Small RNA-sequencing reveals 19 DE blood CholinomiRs in female FMS patients. (**A**) Blood collected in Tempus tubes from female FMS patients and matched controls was used for small RNA-sequencing and miR analysis. (**B**) Volcano plot of altered miR levels in FMS normalized to CTR, and the horizontal line indicates FDA-corrected threshold for DE miRs. Unchanged miRs are in grey, DE miRs are in black, and DE CholinomiRs are in red. (**C**) Counts per million of respective CholinomiRs and log2Fold change compared to CTR. Diameter of dots code for the number of predicted cholinergic targets. (**D**) Assignment of CholinomiRs to blood cell subtypes; magenta indicates absence, and green indicates presence in blood cell type according to [55]. Abbreviations: CholinomiR, cholinergic-targeting microRNA; CPM, counts per million; CTR, healthy controls; DE, differentially expressed; FDA, false discovery rate; FMS, fibromyalgia syndrome; MC, monocyte; miR, microRNA; NK, natural killer cell; NT, neutrophil; RBC, red blood cell.

**Figure 3 cells-11-01276-f003:**
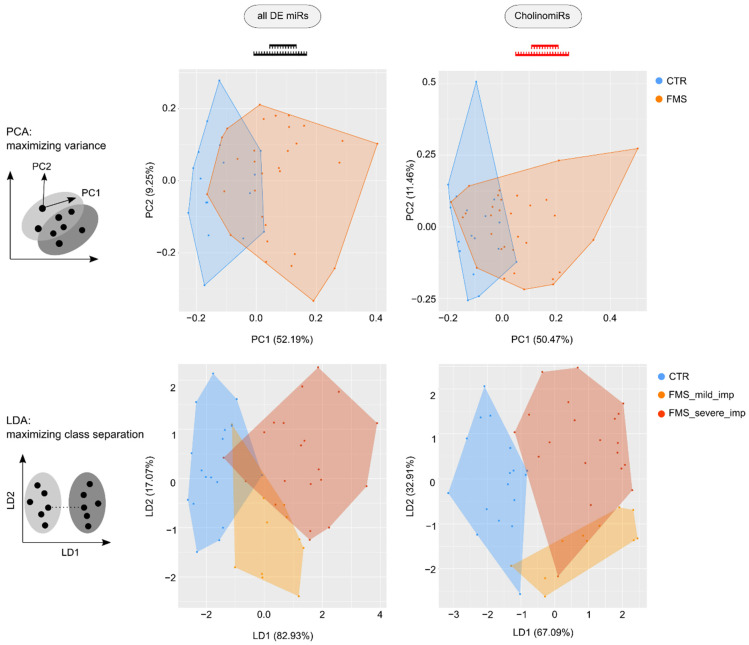
PCA and LDA (principles visualized in grey graphs) for all DE miRs and only CholinomiRs. Although PCA shows only minor improvements in group separation when focusing on CholinomiRs, using LDA with three groups and CholinomiRs leads to high discrimination capacity between healthy controls and FMS patients with mild pain-related daily life impairment (FMS_mild_imp) and severely pain-related daily life impairment (FMS_severe_imp). Abbreviations: CTR, healthy control; DE, differentially expressed; FMS, fibromyalgia syndrome; PCA, principal component analysis.

**Figure 4 cells-11-01276-f004:**
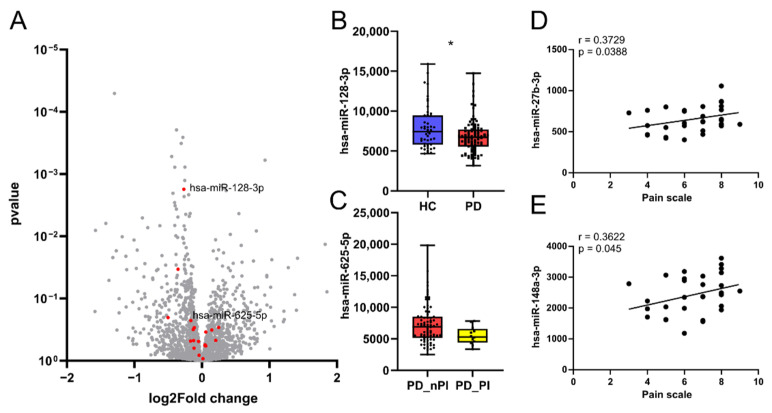
CholinomiRs are related to pain and form a specific FMS signature. Identified CholinomiRs may positively correlate with pain intensity. (**A**) FMS blood levels of hsa-miR-27b-3p (*p* = 0.039; padj = 0.530) and (**B**) hsa-miR.148a-3p (* *p* = 0.045; padj. = 0.563) increased with the mean pain level experienced in the past 6 months, but all 19 CholinomiRs failed to pass post-correction for multiple testing. Spearman correlation test with Holm–Sidak post correction. (**C**) Volcano plot of miRs in PD versus HC with no DE miRs. FMS CholinomiRs present in the PD dataset are marked in red. (**D**,**E**) Only comparing CPM expression levels of FMS CholinomiRs in PD revealed trends for lower hsa-miR-128-3p (*p* = 0.028; padj = 0.364) and within PD patients with pain related impairments an hsa-625-5p downregulation (*p* = 0.052; padj = 0.678). Mann–Whitney U tests with Holm–Sidak post correction. Abbreviations: CPM, counts per million; FMS, fibromyalgia syndrome; HC, healthy control; PD, Parkinson’s disease; PD_nPI, Parkinson’s disease without pain-related daily life impairment; PD_PI, Parkinson’s disease with pain-related daily life impairment.

**Figure 5 cells-11-01276-f005:**
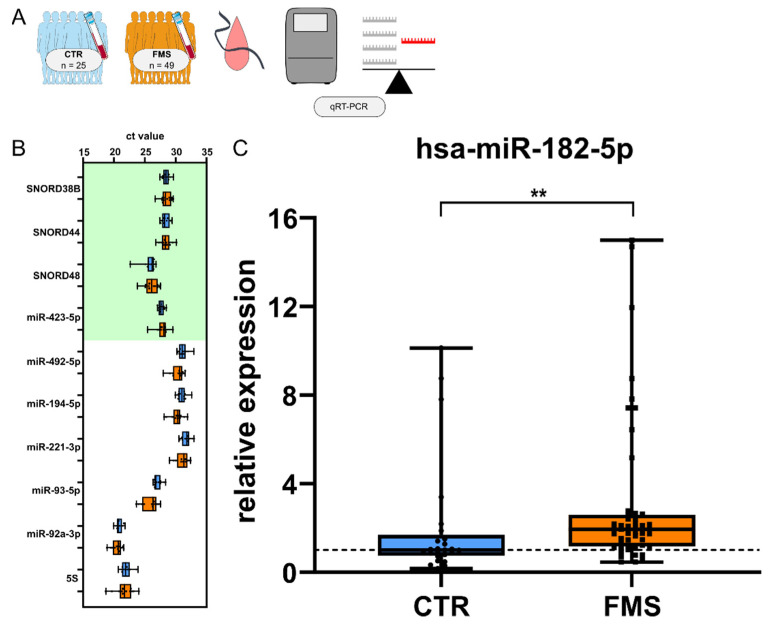
qRT-PCR validation of hsa-miR-182-5p. (**A**) Workflow illustrating cohorts and normalization of hsa-miR182-5p with four unaltered control transcripts. (**B**) Screening of SNORD38B, SNORD44, SNORD48, and hsa-miR-423-5p selected as controls for miR validation and indicated in green. (**C**) hsa-miR-182-5p median expression level is upregulated by twofold in FMS (n = 49), shown as normalized relative expression to CTR (n = 25). The dotted line represents median control expression. Mann–Whitney U Test, ** *p* < 0.01. Box and whiskers plots with min to max. Abbreviations: CTR, healthy controls; FMS, fibromyalgia syndrome; miR, microRNA; qRT-PCR, quantitative real-time PCR.

**Figure 6 cells-11-01276-f006:**
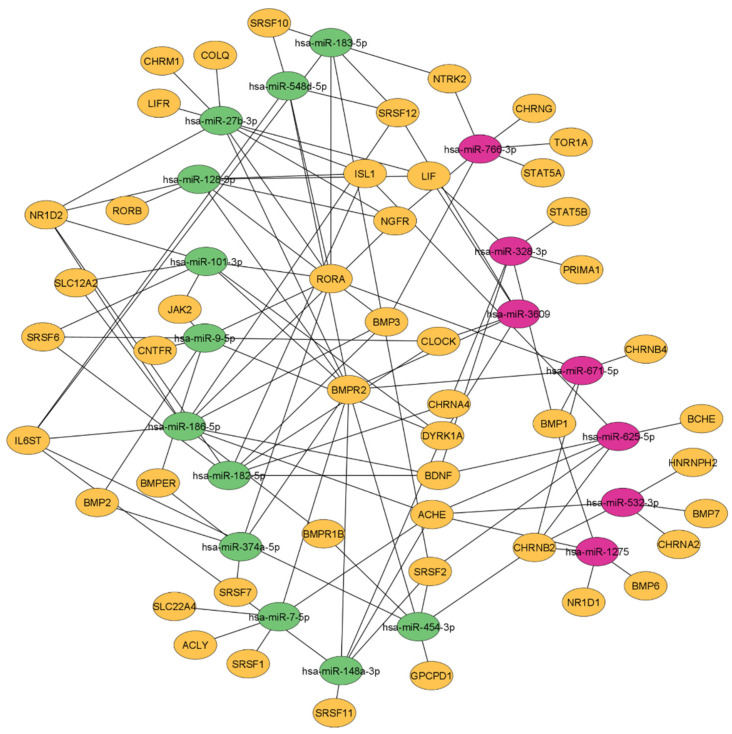
miR-gene interactions. Up-(green) and downregulated (magenta) FMS CholinomiRs are connected to their respective cholinergic target genes (orange) with a total of 47 targets. See comprehensive interaction summary with complete gene names in Appendix A. Abbreviations: miR, microRNA.

**Figure 7 cells-11-01276-f007:**
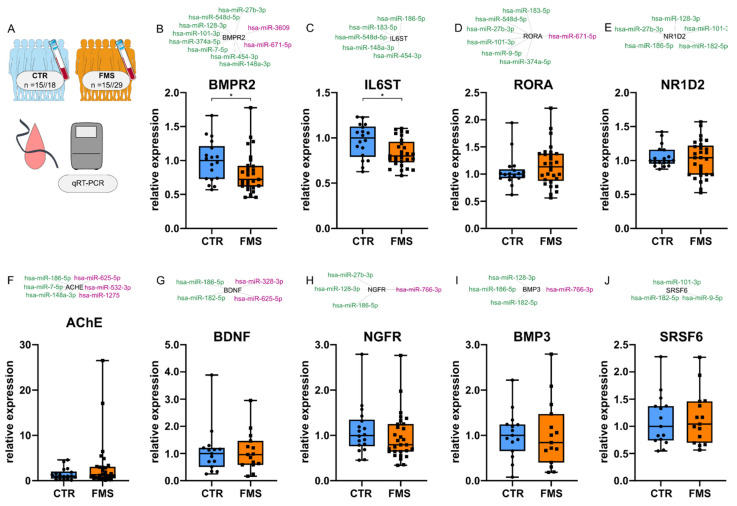
Transcript levels of related cholinergic mRNAs. (**A**) Workflow illustrates cohorts investigated. Global expression of two highly targeted transcripts BMPR2 (**B**) and IL6ST (**C**) was downregulated by 27.8% (* *p* = 0.025) and 19.9% (* *p* = 0.022), whereas RORA (**D**) and NR1D2 (**E**) were not altered in whole blood. ACHE (**F**) and BDNF (**G**), targeted equally by up- and down-regulated CholinomiRs, showed no altered expression, also seen in cholinergic transcripts NGFR (**H**), BMP3 (**I**), and SRSF6 (**J**), which were associated with fewer CholinomiRs. Randomly selected cohort subgroups with n = 29 FMS versus n = 18 CTR (**B**–**F**,**H**) or n = 15 FMS versus n = 15 CTR (**G**,**I**,**J**) were analyzed. Box and whisker plots with min to max. Mann–Whitney U test. Green color depicts upregulated miRs, whereas magenta labels depict downregulated miRs connected to the corresponding mRNA transcript in black.

**Figure 8 cells-11-01276-f008:**
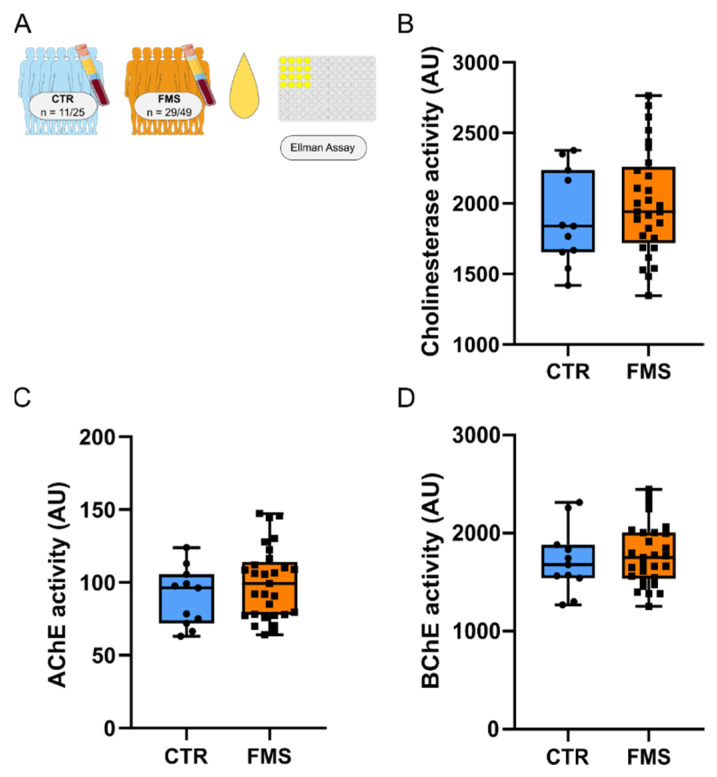
Cholinesterase activity in serum. (**A**) Cohorts and methods. (**B**) Total cholinesterase status in FMS and CTR sera (*p* = 0.401) and subdivision in (**C**) AChE-specific activity (*p* = 0.256) and (**D**) BChE-specific activity (*p* = 0.6224). Mann–Whitney U Test. Abbreviations: AChE, acetylcholinesterase; AU, arbitrary unit; CTR, healthy controls; FMS, fibromyalgia syndrome. Note that there is larger variability in FMS cholinesterase activity compared to that of CTR.

**Table 1 cells-11-01276-t001:** List of applied primer assays.

**SYBR Green Primer**	**Assay Number**	**Company**
hsa-miR-182-5p	YP00206070	Qiagen, Hilden, Germany
hsa-miR-194-5p	YP00204080	Qiagen, Hilden, Germany
hsa-miR-221-3p	YP00204532	Qiagen, Hilden, Germany
hsa-miR-423-5p	YP00205624	Qiagen, Hilden, Germany
hsa-miR-492-5p	YP00204053	Qiagen, Hilden, Germany
hsa-miR-92a-3p	YP00204258	Qiagen, Hilden, Germany
hsa-miR-93-5p	YP00204715	Qiagen, Hilden, Germany
5S	YP00203906	Qiagen, Hilden, Germany
SNORD38B	YP00203901	Qiagen, Hilden, Germany
SNORD44	YP00203902	Qiagen, Hilden, Germany
SNORD48	YP00203903	Qiagen, Hilden, Germany
**Taqman Primer**	**Assay Number**	**Company**
AChE	Hs00241307_m1	Thermo Fisher Scientific, Waltham, MA, USA
BDNF	Hs00380947_m1	Thermo Fisher Scientific, Waltham, MA, USA
BMP3	Hs00609638_m1	Thermo Fisher Scientific, Waltham, MA, USA
BMPR2	Hs00176148_m1	Thermo Fisher Scientific, Waltham, MA, USA
CHRNA4	Hs00181247_m1	Thermo Fisher Scientific, Waltham, MA, USA
CHRNB2	Hs01114010_g1	Thermo Fisher Scientific, Waltham, MA, USA
IL6ST	Hs00174360_m1	Thermo Fisher Scientific, Waltham, MA, USA
NGFR	Hs00609976_m1	Thermo Fisher Scientific, Waltham, MA, USA
NR1D2	Hs00233309_m1	Thermo Fisher Scientific, Waltham, MA, USA
RORA	Hs00536545_m1	Thermo Fisher Scientific, Waltham, MA, USA
SRSF6	Hs05331162_g1	Thermo Fisher Scientific, Waltham, MA, USA

Abbreviations: AChE, acetylcholinesterase; BDNF, brain derived neurotrophic factor; BMP3, bone morphogenetic protein 3; BMPR2, bone morphogenetic protein receptor 2; CHRNA4, cholinergic receptor nicotinic alpha 4 subunit; CHRNB2, cholinergic receptor nicotinic beta 2 subunit; IL6ST, interleukin 6 cytokine family signal transducer; NGFR, nerve growth factor receptor; NR1D2, nuclear receptor subfamily 1 group D member 2; RORA, RAR related orphan receptor A; SNORD, small nucleolar RNA; SRSF6, serine and arginine rich splicing factor 6.

**Table 2 cells-11-01276-t002:** Main clinical parameters of the study cohort that was restricted to women.

Parameter	FMS (n = 49)	FMS-CTR (n = 25)	PD (n = 92)	PD-CTR (n = 47)
Age, median	53	53	56	56
(range)	(25–67)	(24–62)	(34–65)	(31–65)
BMI, median	25	23	n.a	n.a
(range)	(17–40)	(17–42)		
Individual patients on amitriptyline: (Regimen, mg, years)	6	none	none	none
	0-0-1, 10, 10			
	0-0-1, 10, 1			
	0-0-1, 10, 2			
	on demand, 5, 2			
	1-0-0, 10, 1			
	0-0-1, 10, 1			
Graded Chronic Pain Scale; mean pain intensity last 6 months (0–10) (range)	6	0	n.a	n.a
	(2–9)	(0–8)		
Unified Parkinson Disease Rating Scale;	n.a	n.a	0.83	0.46
Pain and other Sensations			(0–4)	(0–3)

Abbreviations: FMS, Fibromyalgia syndrome; FMS-CTR, healthy control cohort for FMS; PD, Parkinson’s disease; healthy control cohort for PD; n.a., not applied.

## Data Availability

Data used in the preparation of this article were obtained from the Parkinson’s Progression Markers Initiative (PPMI) database (www.ppmi-info.org/access-data-specimens/download-data (accessed on 1 March 2021)). For up-to-date information on the study, visit ppmi-info.org.

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
