# Peer review of "Distinct CholinomiR Blood Cell Signature as a Potential Modulator of the Cholinergic System in Women with Fibromyalgia Syndrome"

_cells, 2022, doi:10.3390/cells11081276_

Round 1
Reviewer 1 Report
On line 38 to line 41, you can break it up. It is a long run-on sentence.
On line 45, the sentence is confusing. Risk factors are not necessarily a cause of a symptom such as immune dysregulation. Risk factors are only associated or correlated with a condition. Genetic risk factors can be seen as a biomarker of FMS but this contradicts line 42 about biomarkers. Also, genetic risk factors is vague. It would be helpful to provide specificity to exactly what type of mutations or post-translation changes occurs in the genes.
On line 62 to line 64, the rationale for why mIRs are being investigated is absent. Other than miRs regulating cholinergic systems, there is no other rationale given. Why mIRs and not shRNA or siRNA? With the etiology and pathophysiology of FMS being unknown or misunderstood, there is no clear direction or a priori evidence of why mIR is being measured other than they regulate cholinergic systems. I believe adding in more information such as explaining the projects that led to this line of thinking would give better scientific rigor of the importance of post-translational modification by miRs in the cholinergic system.
On lines 120 to 124, the authors do not state if RNA concentration or RNA quality was assessed for the PD control group and if it was compared to the FMS group. This is important as PCR is a sensitive technique that can result in amplification bias if the starting material are not the same across samples or is partly degraded. It would also increase the transparency of the study if a supplementary table of the RNA concentration and A260/280 ratios were provided.
On lines 125 to 143, the authors uses the collected and purified RNA for RNAseq. Perhaps I am mistaken but RNAseq will sequence all RNA including the undesired RNA species. Was there any particular reason why the RNA could not be purified beforehand to make sure the RNA in downstream analyses were only miR?
On lines 139 to 140, it is appreciated that the authors took a further step and provided an additional criteria for the inclusion of transcripts into the study. Not all genes are transcribed as it is based on need and environmental factors. Clarity if genes or transcripts that fulfilled both criteria or if they fulfilled either criteria were kept would benefit the study.
On lines 415 to 420, it is difficult to compare the pain of FMS to PD patients. I understand this data was collected beforehand and used a completely different test but I would cation the authors to be humble and conservative when comparing pain across conditions. Pain is a field that has very complicated mechanisms and there are many different types and causes of pain. Moreover, it is very subjective. This becomes apparent as in Table 2, a FMS-CTR has a rating of 8 on the pain scale.
Author Response
On line 38 to line 41, you can break it up. It is a long run-on sentence.
We have followed the Reviewer`s suggestion and have broken up the sentence accordingly (now lines 39-42):
“Fibromyalgia syndrome (FMS) is a clinically heterogeneous chronic pain syndrome with a high global prevalence of approximately 2.7% [1,2]. FMS is frequently associated with somatic and mental co-morbidities accompanying the predominantly musculoskeletal pain.”
On line 45, the sentence is confusing. Risk factors are not necessarily a cause of a symptom such as immune dysregulation. Risk factors are only associated or correlated with a condition. Genetic risk factors can be seen as a biomarker of FMS but this contradicts line 42 about biomarkers. Also, genetic risk factors is vague. It would be helpful to provide specificity to exactly what type of mutations or post-translation changes occurs in the genes.
We apologize for not being clear in this point and agree that risk factors do not directly translate to symptoms or diseases. We have re-worded the respective passage (now lines 46-52) in the revised version of our manuscript providing more specific information:
“FMS may aggregate in families and single nucleotide polymorphisms, e.g. in serotonin transporter genes including SLC64A4, the transient receptor potential vanilloid channel 2 gene TRPV2, or mitochondrial genes have been associated with FMS symptoms; however, overall small sample sizes hamper conclusions on the individual patient basis [10,11]. Immune dysregulation based on genetic risk factors e.g. in mitochondrial genes may trigger peripheral neuro-inflammatory processes, which, in turn, could induce central and peripheral pain symptoms;”
On line 62 to line 64, the rationale for why mIRs are being investigated is absent. Other than miRs regulating cholinergic systems, there is no other rationale given. Why mIRs and not shRNA or siRNA? With the etiology and pathophysiology of FMS being unknown or misunderstood, there is no clear direction or a priori evidence of why mIR is being measured other than they regulate cholinergic systems. I believe adding in more information such as explaining the projects that led to this line of thinking would give better scientific rigor of the importance of post-translational modification by miRs in the cholinergic system.
We thank the Reviewer for this valuable comment. Indeed, siRNA are very specialized on disease-related cellular responses upon viral infection. In contrast, miRs regulate a variety of gene transcripts and cellular processes. Hence, we did not consider siRNA in the context of our study. ShRNA represent artificial constructs in order to induce siRNA-mediate modulatory or therapeutic effects. ShRNA does not occur in native human biomaterial as used in our study. To address these issues, we have followed the Reviewer`s suggestion and have re-worded the respective paragraph in the revised version of our manuscript providing more information on the rational for our study design (now lines 70-72):
“Importantly, miRs from blood can be easily obtained, serving as potential biomarkers, but also represent direct modulators in pain conditions [31].”
On lines 120 to 124, the authors do not state if RNA concentration or RNA quality was assessed for the PD control group and if it was compared to the FMS group. This is important as PCR is a sensitive technique that can result in amplification bias if the starting material are not the same across samples or is partly degraded. It would also increase the transparency of the study if a supplementary table of the RNA concentration and A260/280 ratios were provided.
We strongly agree with the Reviewer. Since all subsequent experiments depend on initial good RNA quality and RNA analysis provides valuable input for readers, we now included a Supplementary File 1 with RNA concentrations, A260/280 values, and RNA integrity numbers (RIN). RIN was determined for sequenced samples. The Parkinson's Progression Markers Initiative (PPMI) does not, however, provide concentrations and A260/280 values. Only a small fraction of samples did not achieve a minimum RIN of 7 and exclusion did not alter the results. In addition, RIN gives an indication of how fragmented the RNA is in the sample and miRs (with their small lengths of 22 nts) can be considered fragments, deeming cut-offs based only on RIN values too stringent. To reflect these points, we have extended the respective passages in the revised version of our manuscript (now lines 106-110; lines 125-128):
“The MagMAX™ for Stabilized Blood Tubes RNA Isolation Kit (Thermo Fisher Scientific, Waltham, MA, USA) was used for total RNA extraction according to the manufacturer’s protocol. RNA quality and quantity were assessed with a NanoDrop™ One (Thermo Fisher Scientific, Waltham, MA, USA) and RNA was stored at -80°C. Sequenced samples were analyzed via Agilent 2100 Bioanalyzer system (Agilent Technologies, Santa Clara, CA, USA; see details for RNA quality in Supplementary File 1).”
“These were incubated at room temperature (18–25°C) for 24 h before final storage at -80°C. RNA was extracted using the PAXgene blood miRNA kit protocol (Qiagen, Hilden, Germany; see details for RNA quality in Supplementary File 1).”
On lines 125 to 143, the authors uses the collected and purified RNA for RNAseq. Perhaps I am mistaken but RNAseq will sequence all RNA including the undesired RNA species. Was there any particular reason why the RNA could not be purified beforehand to make sure the RNA in downstream analyses were only miR?
We have collected blood samples in Tempus tubes (FMS patients and healthy controls) and have integrated data derived from PAX tubes (PD patients and respective controls). Both share the advantage of including small RNA and mRNA within one sample in a snapshot manner. For subsequent sequencing, the NEBNext Small RNA Library Prep Set for Illumina was used which contains a size selection step after PCR to ensure that only small RNAs up to 140 nt length will be sequenced. We refer to the instruction manual of the NEB-E7560S (NEBNext® Multiplex Small RNA Library Prep Set for Illumina® Set 1,Set 2, Index Primers 1–48 and Multiplex Compatible) and have added this information in the revised version of our manuscript (now lines 136-140):
“For this purpose, 300 ng RNA per sample served for library preparation using the NEB-Next® Multiplex Small RNA Library Prep Kit for Illumina® (Index Primers 1-48; NEB-E7560S; New England Biolabs, Ipswich, MA, USA) at the National Center for Genomic Technologies at the Hebrew University of Jerusalem. According to the information provided by the manufacturer, this kit contains a size selection step after PCR to ensure that only small RNAs up to 140 nt length will be sequenced.”
On lines 139 to 140, it is appreciated that the authors took a further step and provided an additional criteria for the inclusion of transcripts into the study. Not all genes are transcribed as it is based on need and environmental factors. Clarity if genes or transcripts that fulfilled both criteria or if they fulfilled either criteria were kept would benefit the study.
We thank the Reviewer for this critical and important question. Since we did not perform mRNA sequencing of Tempus tube RNA-derived samples, we could perform expression analysis only via qRT-PCR. However, to address this issue, we extracted RNA expression information provided by the Human Protein Atlas (HPA) project. We included the normalized transcripts per million (TPM) values for 18 immune cell subtypes and total peripheral blood mononuclear cells of all 47 identified miR targets. Please note that this dataset does not include RNA sequencing data of red blood cells, which also might contribute substantially to Tempus tube-derived RNA fractions. Further, the HPA dataset does not allow conclusions on aberrant gene expression in FMS blood. We have now included this information as a new Supplementary File 3 in the revised version of our manuscript and believe that it will allow readers to gain a more comprehensive view on potential miR-gene transcript interactions in given immune cell subtypes. We have also extended the respective passage in the revised version our manuscript (now lines 389-400):
“We compared our candidate gene list with an online mRNA sequencing dataset derived from healthy donor blood RNA, from the Human Protein Atlas (HPA) project to assess the expression profile of these miR targets across single immune cell subtypes (see Supplementary File 3). With 61.7% (29/47 genes), natural killer (NK) cells expressed the highest number of mRNA targets, followed by neutrophils with 57.5% (27/47), while myeloid dendritic cells showed the lowest numbers of detected mRNA targets at 44.7% (21/47). Among the top-targeted transcripts, BMPR2, IL6ST, RORA, and NR1D2 were expressed in almost all cell types, while ACHE and BDNF expression was not detected in this dataset. However, red blood cells were not included in the HPA dataset, which represents a large fraction of Tempus tube derived-RNA and no disease-associated immune cell data is included.”
On lines 415 to 420, it is difficult to compare the pain of FMS to PD patients. I understand this data was collected beforehand and used a completely different test but I would cation the authors to be humble and conservative when comparing pain across conditions. Pain is a field that has very complicated mechanisms and there are many different types and causes of pain. Moreover, it is very subjective. This becomes apparent as in Table 2, a FMS-CTR has a rating of 8 on the pain scale.
We thank the Reviewer and have re-worded the respective paragraph in the revised version of our manuscript (now lines 477-483):
“While we do not define a “unique” FMS CholinomiR signature, comparisons with healthy and disease controls may indicate distinct small RNA-mediated mechanisms of pain in FMS and PD. Pain phenotypes, perception, and methodological assessment of pain substantially vary across individuals and between diseases. The exact roles of these candidate miRs in different pain conditions would need to be investigated in future studies with respective patient cohorts.”
Submission Date
28 February 2022
Date of this review
11 Mar 2022 18:18:33

Reviewer 2 Report
The manuscript is clear and well written, although some minor typos.
I would suggest to explore and describe how the female hormone can have an impact on the cholinergic system and if they found any relation according of the target genes.
According to the literature around Cholinergic system and inflammation I strongly recommend to consider this link in their introduction and then, in the discussion, explain this role linked with their results.
Figure 6: Although the explanation in the results section is clear, I would prefer to have a table where the connections are listed. I suggest to describe if the connection is new or already present in literature.
Fig. 7: The result of the IL6 downregulation is very interesting and it can be easily correlated with the previous results showing the control of cholinergic system on the proinflammatory cytokines. I suggest to confirm this result with ELISA assay of blood samples used in this study.
Figure 8: Western blot of AChE and BuChE could be a nice addition together with the enzymatic activity. Please add.
Author Response
The manuscript is clear and well written, although some minor typos.
Thank you, we have proof-read the manuscript again and have eliminated typos.
I would suggest to explore and describe how the female hormone can have an impact on the cholinergic system and if they found any relation according of the target genes.
We are aware of sex-specific differences in cholinergic signaling shown in neurons [1] and assume downstream effects on the organism systemically and also diverse signaling in immune cells. In this study, we did not include men, precluding subdivision of the cohort for determination of sex-specific differences of the cholinergic system in general or specifically in FMS. We will use these initial findings to address this important issue in future studies.
According to the literature around Cholinergic system and inflammation I strongly recommend to consider this link in their introduction and then, in the discussion, explain this role linked with their results.
We have followed the Reviewer`s suggestion and have re-worded the respective passages in the Introduction (please see lines 62-65), Discussion (please see lines 535-536), and Conclusion (please see lines 585-587) sections of our revised manuscript:
Introduction: “Together, cholinergic mechanisms play an important role in the regulation of cytokines and antibody production [23,24], thereby modulating inflammatory processes with potential implications for FMS pathophysiology and severity [25,26].”
Discussion: “Taken together, CholinomiRs may shift inflammatory processes via modulation of the systemic cholinergic system.”
Conclusion: “While the exact pathophysiological mechanisms remain unclear, CholinomiRs emerge as important posttranscriptional regulators, which may mediate pathological pro-inflammatory or compensatory effects in immune cells of FMS patients.”
Figure 6: Although the explanation in the results section is clear, I would prefer to have a table where the connections are listed. I suggest to describe if the connection is new or already present in literature.
A list of all connections is indeed presented in (now) Supplementary File 2. The previously published algorithm (’mirNet’) [1] integrates already known connections from ten miR-gene interaction datasets. To decrease false-positive connections, only cumulative evidence (each database “hit” adding 1 point, except strong evidence in miRTarBase giving 10.5 points) was used. Only connections with a cumulative value of 7or more were then considered valid. We have included the respective information in the Methods section of our revised manuscript (please see lines 367-372):
“The previously published miR-targeting graph database ‘miRNet’ allowed in-depth exploration of the CholinomiR-gene transcript network by integrating publicly available validated and predicted miR-gene interaction datasets [40]. Our analysis revealed 47 potential cholinergic targets and 118 miR-mRNA interactions (Figure 6) with upregulated cholinergic miRs forming 80 connections compared to 38 connections of downregulated miRs (for details see Supplementary File 2).”
Fig. 7: The result of the IL6 downregulation is very interesting and it can be easily correlated with the previous results showing the control of cholinergic system on the proinflammatory cytokines. I suggest to confirm this result with ELISA assay of blood samples used in this study.
Thank you for this comment. Indeed, we found a downregulation of IL6ST (gp130) in FMS compared to controls. Unfortunately, we have collected serum, but not whole blood samples for protein analysis (only Tempus tubes for RNA extraction) from our patients. Hence, we cannot address this issue properly, since ELISA of serum would not deliver comprehensive data on this membrane receptor mainly located within immune cells.
Figure 8: Western blot of AChE and BuChE could be a nice addition together with the enzymatic activity. Please add.
Please note that both the AChE/BUChE activity and blood RNA levels were not altered in our FMS cohort. Therefore, the capacity for production and the longevity of these enzymes appeared to be unchanged. Moreover, protein blots would detect antibody binding also against catalytically inactive molecules, eliciting one more reason for potential error. For all of these reasons, we assume the likelihood of relevantly altered protein expression levels to be found using Western blot as small and would respectfully refrain from adding this experiment.
Submission Date
28 February 2022
Date of this review
28 Mar 2022 17:07:09
References:
- Lobentanzer, S.; Hanin, G.; Klein, J.; Soreq, H. Integrative transcriptomics reveals sexually dimorphic control of the cholinergic/neurokine interface in schizophrenia and bipolar disorder. Cell Rep. 2019, 29, 764-777. e765.

Reviewer 3 Report
This manuscript sets out to identify miR molecules involved in cholinergic pathways that are differentially expressed in Fibromyalgia syndrome (FMS) and furthermore unique to FMS compared to a control condition that also exhibits chronic pain symptoms, Parkinson's disease (PD).
The authors began by appropriately identifying female patients, whom are most affected by FMS, as well as healthy controls for small RNA sequencing analysis from blood samples. Differential miR targets were identified, particularly in FMS patients with high daily life impairment. Several of these miR molecules were found to be unique to FMS when compared to PD data. Two of these molecules were found to roughly correlate with patient pain scales and were unique to FMS (hsa-miR-27b-3p and hsa-miR.148a-3p).
One miR, hsa-miR-182-5p, was found to exhibit a 2-fold upregulation in FMS patients vs the control group, as confirmed by qRT-PCR. Furthermore, mRNA transcript levels of several genes involved in cholinergic pathways were assessed by qRT-PCR, identifying that BMPR2 and IL6ST were significantly down-regulated in FMS patients whereas other genes remained unchanged.
Lastly, cholinesterase activity was measured and found to be not different between control and FMS patients.
The discussion successfully addresses the various miR identified in the context of the literature, their targets and implications ranging from neuronal oxidative stress to pro-inflammatory pathways possibly tied to circadian rhythm regulation, leaving many avenues for further investigation.
The study is generally well conceived and put together and significant differences were found albeit very modest ones.
There are some critical details missing:
1) While the authors have been very careful to use appropriate blood collection methods for the isolation of stable RNA, no RNA integrity analysis was reported. These details and the general outcome need to be included in the methodology. RNA integrity numbers should be a minimum of 7 or higher.
2) For the qRT-PCR the methodology is lacking in detail and rigor. What were primer efficiencies and linear range of the assay? Did the experimental Ct values fall within this range? Commercially purchased primer sets still require experimental validation.
3) What was the rationale for selecting the SNORD38b, SNORD44 and SNORD48 as endogenous controls? This needs to be explained.
4) What about RPL13A - why was it used as the endogenous control for the mRNA analysis?
5) Please expand on the controversial role of BDNF in FMS and how does the new data tie into this literature?
Author Response
This manuscript sets out to identify miR molecules involved in cholinergic pathways that are differentially expressed in Fibromyalgia syndrome (FMS) and furthermore unique to FMS compared to a control condition that also exhibits chronic pain symptoms, Parkinson's disease (PD).
The authors began by appropriately identifying female patients, whom are most affected by FMS, as well as healthy controls for small RNA sequencing analysis from blood samples. Differential miR targets were identified, particularly in FMS patients with high daily life impairment. Several of these miR molecules were found to be unique to FMS when compared to PD data. Two of these molecules were found to roughly correlate with patient pain scales and were unique to FMS (hsa-miR-27b-3p and hsa-miR.148a-3p).
One miR, hsa-miR-182-5p, was found to exhibit a 2-fold upregulation in FMS patients vs the control group, as confirmed by qRT-PCR. Furthermore, mRNA transcript levels of several genes involved in cholinergic pathways were assessed by qRT-PCR, identifying that BMPR2 and IL6ST were significantly down-regulated in FMS patients whereas other genes remained unchanged.
Lastly, cholinesterase activity was measured and found to be not different between control and FMS patients.
The discussion successfully addresses the various miR identified in the context of the literature, their targets and implications ranging from neuronal oxidative stress to pro-inflammatory pathways possibly tied to circadian rhythm regulation, leaving many avenues for further investigation.
The study is generally well conceived and put together and significant differences were found albeit very modest ones.
There are some critical details missing:
1) While the authors have been very careful to use appropriate blood collection methods for the isolation of stable RNA, no RNA integrity analysis was reported. These details and the general outcome need to be included in the methodology. RNA integrity numbers should be a minimum of 7 or higher.
We strongly agree with the Reviewer and please also see our reply to question four of Reviewer 1. Since all subsequent experiments depend on initially good RNA quality and as RNA analysis provides valuable input for readers, we now included a Supplementary File 1 with RNA concentrations, A260/280 values, and RNA integrity numbers (RIN). RIN was determined for sequenced samples. The Parkinson's Progression Markers Initiative (PPMI) does, however, not provide concentrations and A260/280 values. Only a small fraction of samples did not achieve a minimum RIN of 7 and exclusion did not alter the results. In addition, RIN gives an indication of how fragmented the RNA is in the sample and miRs (with their small lengths of 22 nts) can be considered fragments, deeming cut-offs based only on RIN values too stringent. We have extended the respective passages in the revised version of our manuscript (now lines 106-110; lines 125-128).
“The MagMAX™ for Stabilized Blood Tubes RNA Isolation Kit (Thermo Fisher Scientific, Waltham, MA, USA) was used for total RNA extraction according to the manufacturer’s protocol. RNA quality and quantity were assessed with a NanoDrop™ One (Thermo Fisher Scientific, Waltham, MA, USA) and RNA was stored at -80°C.Sequenced samples were analyzed via Agilent 2100 Bioanalyzer system (Agilent Tech-nologies, Santa Clara, CA, USA; see details for RNA quality in Supplementary File 1).”
“These were incubated at room temperature (18–25°C) for 24 h before final storage at -80°C. RNA was extracted using the PAXgene blood miRNA kit protocol (Qiagen, Hilden, Germany; see details for RNA quality in Supplementary File 1).”
2) For the qRT-PCR the methodology is lacking in detail and rigor. What were primer efficiencies and linear range of the assay? Did the experimental Ct values fall within this range? Commercially purchased primer sets still require experimental validation.
We agree that in case of e.g. custom-designed primer sets specificity and efficiency experiments are mandatory. However, for pre-designed, wet-bench validated assays, this is usually unnecessary and primer specificity is guaranteed by the manufacturing company (here Qiagen for small RNAs and Thermo Fisher Scientific for mRNA). Since SYBR green assays are considered less specific compared to Taqman assays, we ensured the functionality and specificity of Qiagen primers via melting curve step assessment for miR qRT-PCR. We have added this information to the Methods section of our revised manuscript (now lines 194-195):
“Functionality and specificity of SYBR green based primers were checked via melting curve step assessment.”
3) What was the rationale for selecting the SNORD38b, SNORD44 and SNORD48 as endogenous controls? This needs to be explained.
We are happy to discuss the selection of endogenous controls for miR qRT-PCR quantification. We included potential endogenous controls from literature and derived from analysis of the small RNA Seq data. From literature, we included SNORD38B, SNORD44, SNORD48, 5S RNA, and hsa-miR-221-3p). Hsa-miR-16 and U6 RNA were not considered, due to contradictory data regarding their stability and potential deregulation in diseases. We subjected the normalized counts from sequencing (excluding miRs with 0 expression in any of the samples) to NormFinder. Afterwards, miRs were selected on their inter- and intra-group variability, together with a minimum mean read expression level as cut-off (min 1000 reads), which led to selection of hsa-miR-942-5p, hsa-miR-194-5p, miR-93-5p, miR92a-3p, and miR423-5p. These candidates were analyzed via qRT-PCR (see Figure 5A). As a single small RNA transcript cannot act as a bona-fide endogenous control, we aimed to include four out of these candidates and normalize against their geometric mean. Based on the group differences, range and standard deviation of Ct values for these candidates, we decided on SNORD38B, SNORD44, SNORD48, and hsa-miR-423-5p. For clarification, we included the following paragraph in the Methods section of our revised manuscript (now lines 182-193):
“Following a dual selection approach, we included ten transcripts as potential endogenous controls for small RNAs. SNORD38B, SNORD44, SNORD48, 5S RNA, and hsa-miR-221-3p were selected from literature. Further, miRs with a mean base read level of >1000 reads within the small RNA sequencing dataset were selected by their intra- and intergroup stability, assessed by NormFinder [45], identifying hsa-miR-942-5p, hsa-miR-194-5p, miR-93-5p, miR92a-3p, and miR423-5p as candidates. Twelve random CTR and FMS samples were used for validation via qRT-PCR. SNORD38B, SNORD44, SNORD48, and miR-423-5p were selected as suitable endogenous controls, based on Ct intergroup comparability and standard deviation and range across all samples. To minimize the risk of potential skewed normalization by only one endogenous control, we normalized against the combined geometric mean of these four transcripts.”
4) What about RPL13A - why was it used as the endogenous control for the mRNA analysis?
We agree that RPL13A is one of several valid candidates for a housekeeping gene/endogenous control for relative mRNA quantification via qRT-PCR [1]. In our laboratory, we successfully use RPL13A as an endogenous control for duplex-based assays and had initially chosen RPL13A after comparison with potential other endogenous controls such as ACTB, B2M, GAPDH, HMBS, and TBP in several human cell culture lines. For the current study, we tested RPL13A suitability via qRT-PCR with 18 controls versus 29 FMS patients. Both groups showed normal distribution (D´Agostino & Pearson omnibus normality test with p = 0.318/ p = 0.311 and Shapiro-Wil normality test with p = 0.700/ p = 0.278) and highly comparable and stable median (21.00 versus 20.93) and mean (20.99 versus 21.06) Ct values. In two-tailed unpaired t-test CTR versus the FMS group were not different (p = 0.626). Hence, we are confident that RPL13A is a suitable endogenous control to be used in our study.
5) Please expand on the controversial role of BDNF in FMS and how does the new data tie into this literature?
We agree with the Reviewer that current data on BDNF in FMS is contradictory. Unfortunately, our study design does not allow robust conclusions in either way and we would refrain from adding speculations not strongly grounded on data. However, we have followed the Reviewer`s suggestion, and have extended the Discussion section of our revised manuscript referring to BDNF (now lines 485-492).
“This miR targets e.g. BDNF, which predominantly derives from the central nervous system, but is also secreted by activated immune cells and acts as a positive regulator of survival and differentiation of PNS and CNS neurons [69]. Reduced blood levels of BDNF have been reported in major depression (MD) and depressed mood is a frequent co-morbidity in FMS [3,70]. However, contrarily to MD, some studies reported increased serum BDNF levels in FMS patients [71,72], while others did not find any deregulation [73], which obscures its role in FMS pathology. Our study exclusively focuses on blood RNA expression, leaving our findings hinting towards a potential lack of BDNF deregulation speculative.”
Submission Date
28 February 2022
Date of this review
28 Mar 2022 21:52:29
References:
- De Jonge, H.J.; Fehrmann, R.S.; de Bont, E.S.; Hofstra, R.M.; Gerbens, F.; Kamps, W.A.; de Vries, E.G.; van der Zee, A.G.; te Meerman, G.J.; ter Elst, A. Evidence based selection of housekeeping genes. PloS one 2007, 2, e898.
